# Epoxidized Soybean Oil Toughened Poly(lactic acid)/Lignin-g-Poly(lauryl methacrylate) Bio-Composite Films with Potential Food Packaging Application

**DOI:** 10.3390/polym16142025

**Published:** 2024-07-16

**Authors:** Yingxin Zhou, Kang Shi, Guoshuai Liu, Hui Sun, Yunxuan Weng

**Affiliations:** 1School of Light Industry Science and Engineering, Beijing Technology and Business University, Beijing 100048, China; zhouyingxin@btbu.edu.cn (Y.Z.); dushuer6247287@163.com (K.S.); 18838935071@163.com (G.L.); 2Beijing Key Laboratory of Quality Evaluation Technology for Hygiene and Safety of Plastics, Beijing Technology and Business University, Beijing 100048, China

**Keywords:** poly (lactic acid), lignin, epoxy groups, food packaging

## Abstract

The application of lignin as a filler for poly (lactic acid) (PLA) is limited by their poor interfacial adhesion. To address this challenge, lignin-graft-poly(lauryl methacrylate) (LG-g-PLMA) was first blended with poly (lactic acid), and then epoxidized soybean oil (ESO) was also added to prepare PLA/LG-g-PLMA/ESO composite, which was subsequently hot pressed to prepare the composite films. The effect of ESO as a plasticizer on the thermal, mechanical, and rheological properties, as well as the fracture surface morphology of the PLA/LG-g-PLMA composite films, were investigated. It was found that the compatibility and toughness of the composites were improved by the addition of ESO. The elongation at break of the composites with an ESO content of 5 phr was increased from 5.6% to 104.6%, and the tensile toughness was increased from 4.1 MJ/m^3^ to 44.7 MJ/m^3^, as compared with the PLA/LG-g-PLMA composite without ESO addition. The toughening effect of ESO on composites is generally attributed to the plasticization effect of ESO, and the interaction between the epoxy groups of ESO and the terminal carboxyl groups of PLA. Furthermore, PLA/LG-g-PLMA/ESO composite films exhibited excellent UV barrier properties and an overall migration value below the permitted limit (10 mg/dm^2^), indicating that the thus-prepared biocomposite films might potentially be applied to environmentally friendly food packaging.

## 1. Introduction

Recently, concerns about environmental pollution and regulations enforcement related to sustainable development have stimulated an increase in environmental protection efforts [1]. In this case, biodegradable polymers, especially biomass, have gained growing research attention [2,3].

Polylactic acid (PLA) is a bio-based polyester and can be obtained entirely from renewable resources [4,5]. PLA has received widespread attention for its excellent machinability, biocompatibility, and biodegradability, and it is considered to be a promising substitute to traditional petroleum-based polymers [6]. PLA has currently found application in packaging, textile, agriculture, and the medical industry [7,8,9]. However, brittleness, slow crystallization rate, and high cost are several drawbacks that limit its application [10]. In view of this, low-cost fillers are often used to reduce the cost, and plasticizers have been employed to the improvement of toughness [11,12,13].

Among the materials from renewable resources used as fillers for PLA, lignin has shown good performance with the advantages of abundant reserves and wide availability. With abundant active functional groups and highly branched polyphenolic aromatic structures, lignin can be chemically modified for polarity adjustment [14,15,16,17]. Lignin has been incorporated to PLA in an effort to reduce cost and improve heat resistance and barrier properties [18,19]. Nevertheless, blending lignin with PLA has remained a challenge because lignin tends to aggregate in PLA matrix, leading to deteriorated interfacial adhesion, and thereby decreased mechanical properties [20,21].

In this regard, chemical modification of lignin has been explored for the purpose of improving its adhesion with PLA [17,18,22,23]. The graft modification of lignin by ordinary free radical polymerization, such as chemical grafting of lignin with lauryl methacrylate (LMA) and tetrahydrofurfuryl methacrylate (THFMA), considerably improved the interfacial compatibility and mechanical properties of PLA/grafted lignin composites [24]. Ring-opening polymerization was also reported to graft lactide onto lignin nanoparticles (LNP); the LNP was found uniformly dispersed in PLA matrix without aggregation. Using this strategy the elongation at break was significantly increased, and the composite films showed stronger UV absorption properties and antioxidant properties [19].

Moreover, in order to improve its toughness, PLA was also plasticized via copolymerization or compounding with flexible polymers (e.g., PBAT, PBSA, ABS, PCL) and the use of oligomer and low-molecular additives, etc. [25]. With regard to compounding with flexible polymers, compatibility needs to be considered. The introduction of oligomeric and low-molecular-weight plasticizers is a much more convenient, effective, and low-cost way to improve the toughness of PLA [26]. Bio-based plasticizers have been widely used to improve the processability and flexibility of PLA composites owing to their renewability and environmental friendliness [27,28,29,30]. In particular, epoxidized soybean oil (ESO) was investigated because its epoxy group can readily be bonded to functional group of other polymers via covalent or hydrogen bonds [31]. Specifically, ESO has been extensively employed to the modification of PLA, where its epoxy group can react with the terminal carboxyl group of PLA, displaying an effect similar to that of an active plasticizer. Xiong et al. incorporated ESO to PLA/maleic anhydride (MA)-grafted starch composites, and noted that ESO, as a plasticizer, considerably elevated the elongation at break, and enhanced the compatibility of PLA/starch [32]. Arkadiusz et al. plasticized PLA using acrylic epoxidized soybean oil, which resulted in films with very high toughness and elongation at break of almost 800% [33]. In addition, it was also found that the epoxy functional group in ESO can react with the hydroxyl group of lignin to promote the dispersion of lignin in polyester by reducing self-aggregation, thereby augmenting the compatibility between polyester and lignin [34]. Studies have also found that ESO does not easily migrate into food [35]. Therefore, ESO-plasticized PLA/lignin was also considered a promising green and flexible food packaging material [36].

In our previous work [37], we reported the preparation of lignin-graft-poly(lauryl methacrylate) (LG-g-PLMA) by ordinary free radical polymerization. The thus-prepared graft polymer was subsequently melt compounded with PLA. It was noted that the compatibility of the copolymers with PLA was improved. In virtue of the presence of soft LMA segments in LG-g-PLMA, the elongation at break of the composites were 1.4 times higher than that of pure PLA [37]. However, for food packaging applications, modification of PLA/LG-g-PLMA composites remains a challenge with respect to further improvement in compatibility and toughness. In this respect, evaluation on safety for food contact is another concern. To this end, bio-based ESO was incorporated as a plasticizer into PLA/LG-g-PLMA composites in this work. Our hypothesis was that the epoxy group of ESO interacted with the terminal carboxyl group of PLA, as shown in Figure 1, with the expectation that ESO could noticeably improve the compatibility and toughness of the composites, and the overall migration comply with the relevant regulations. The focus of the present work lies in the effect of ESO on the thermal, mechanical properties, morphology, as well as the overall migration of the composites, to reveal the toughening mechanism and demonstrate the potential of the composites for application in food packaging.

## 2. Materials and Methods

### 2.1. Materials

Poly (lactic acid) (PLA 4032D, density = 1.240 g/cm^3^, melt flow index 7.0 g/10 min (210 °C, 2.16 kg)) was supplied from Nature Works LLC, Blair, NE, USA.

Lignin (purity 80%) was obtained from Shandong Longlive Biotechnology Co., Ltd., Qingdao, China.

Epoxidized soybean oil (ESO, AR, epoxy value ≥ 6%), lauryl methacrylate (LMA, AR) (stabilized, 96%), and dimethyl sulfoxide (DMSO, AR) were obtained from Macklin Chemical Reagent Co., Ltd., Shanghai, China.

Calcium chloride (CaCl_2_, AR), hydrogen peroxide (H_2_O_2_, AR), and hydrochloric acid (HCl) were acquired from Fuchen Chemical Reagent Co., Ltd., Tianjin, China.

### 2.2. Preparation of LG-g-PLMA

LG-g-PLMA was synthesized using a procedure described in our previous work [37]. In a typical procedure, under a nitrogen atmosphere, 1.0 g of CaCl_2_ and 20 mL of DMSO were added to a flask and stirred until CaCl_2_ was completely resolved. Subsequently, 2.0 g of lignin was added, and the reaction temperature was raised to 50 °C and stirred for 30 min until the dissolution of lignin. At last, 4.0 g of LMA and 1 mL of hydrogen peroxide were added to the flask, and the reactants were stirred for 6 h.

At the end of this reaction, the solution was added dropwise to diluted HCl (pH = 3) to precipitate. Deionized water was used to rinse and filter the precipitate.

Afterwards, the finished product LG-g-PLMA was vacuum dried. The grafting efficiency of the copolymer was determined to be 80.6% according to the following equation.
(1)$$G(\%)=\frac{W_{2}-W_{1}}{W_{0}}\times 100\%$$
where *W*_2_ is the weight of the graft copolymer, *W*_1_ is the weight of the LG used, and *W*_0_ is the weight of the monomer.

### 2.3. Preparation of PLA/LG-g-PLMA/ESO Composite

PLA was vacuum dried for 8 h at 80 °C. LG-g-PLMA was vacuum dried for 24 h at 60 °C. Afterward, PLA/LG-g-PLMA /ESO composites were prepared with an internal mixer (XSS-300, Shanghai Kechuang, Shanghai, China) at 185 °C and 60 rpm for 7 min. The sample formulations are provided in Table 1. The proportion of the ESO was relative to the total amount of PLA and LG-g-PLMA.

### 2.4. Preparation of PLA/LG-g-PLMA/ESO Composite Film

PLA/LG-g-PLMA composite film was prepared by hot pressing at 185 °C and 6.5 MPa with a hydraulic press machine (LP-S-50, Lab-Tech, Beijing, China), with a cooling time of 10 min, and the load remained unchanged during the cooling period. The film thickness was 150–220 µm.

### 2.5. Characterization

(1)Fourier Transform Infrared Spectroscopy (FT-IR)

Fourier transform infrared (FT-IR) spectra were obtained using a Thermo Scientific instrument Nicolet iS10 in the attenuated total reflection (ATR) mode with wavenumbers range between 4000 and 400 cm^−1^, scanning frequency of 64, and 4 cm^−1^ resolution.

(2)Differential Scanning Calorimetry (DSC)

Samples (approximately 3–5 mg) were weighed. Differential scanning calorimetry (DSC) was conducted with a TA instrument DSC Q100 (Newcastle, DE, USA) with a heating rate of 10 °C/min and a gas flow rate of 50 mL/min from 30–200 °C. The degree of crystallinity (*X*c) for each sample was calculated by the following equation.
(2)$$X_{c}=\frac{\Delta H_{m}-\Delta H_{cc}}{\varphi_{PLA}\times \Delta H_{100\%PLA}}\times 100\%$$
where Δ*H_m_* is the melting enthalpy, Δ*H_m_* is the cold crystallization enthalpy, *φ_PLA_* is the mass fraction of *PLLA*, and Δ*H*_100%*PLLA*_ is the melting enthalpy of pure *PLLA* (Δ*H*_100%*PLLA*_ = 93.7 J/g) [38].

(3)Mechanical Properties

The tensile properties of the samples were measured on a universal testing machine UTM-1422 in accordance with the China national standard GB/T 1040.3-2006 at a rate of 3 mm/min [39]. The length of the tensile specimens was 100 mm, and the width was 10 mm. At least five specimens from each batch were tested to obtain an average value.

(4)Thermogravimetric Analysis

Samples (approximately 5 mg) were weighed. Determination of thermal stability of the samples was performed on a Hitachi instrument STA7200 (Tokyo, Japan) with a heating rate of 20 °C/min and a gas flow of 50 mL/min under a nitrogen atmosphere. Experiments were carried out in the temperature range from room temperature to 700 °C.

(5)
*Rheological Properties*


The rheological test was performed by a Thermo Scientific instrument MARS (Waltham, MA, USA) at 190 °C with an angular frequency (ω) scanning range of 0.1–100 rad/s and a deformation of 1% in the linear viscoelastic region, under an air atmosphere. The parallel plate was 20 mm in diameter with a plate gap of 1 mm.

(6)Optical Property

The optical properties of samples were determined on a Shimadzu equipment UV-3600 (Kyoto, Japan) in the wavelength interval between 200 and 800 nm.

(7)Scanning Electron Microscopy

The microscopic morphology of the cryo-fractured and tensile fracture surfaces was investigated by Thermo Scientific instrument Quattro S (Waltham, MA, USA) with 4000 times magnification. Each specimen was sputtered with a thin layer of gold.

(8)Oxygen transmission rate

The gas transmission rate of samples was determined by Labthink gas permeability tester VAC-V2 (Jinan, China) according to ASTM D3985 [40], at 23 °C with 30% relative humidity. The film samples (10 cm^2^) were deposited in the penetration chamber of the tester.

(9)Overall migration

In the overall migration tests, composite samples (120 cm^2^) were immersed in 200 mL of water, 3% vol % acetic acid in water (acidic food simulants), 10 vol % ethanol in water (aqueous food simulants), 50 vol % ethanol in water (milk and alcohol beverage simulants), and 90 vol % ethanol in water (fatty food simulants) respectively, and kept for 10 d at 40 °C according to the China National Standard GB 31604.1 [41]. At the end of immersion, the samples were removed, and the simulants evaporated. Overall migration values were derived by the calculation of the mass differences between samples before and after treatment. 

## 3. Results and Discussion

### 3.1. Structural Characterization of PLA/LG-g-PLMA/ESO

The chemical structures of the composites were characterized by FT-IR before and after the addition of ESO, as shown in Figure 1. Peaks appearing at 1738 and 1743 cm^−1^ were attributed to C=O stretching vibrations of ESO and PLA/LG-g-PLMA, respectively. Peaks observed at 1178 and 1077 cm^−1^ originated from the C-O stretching vibrations on the aliphatic chain of PLA; two absorption bands attributed to PLA were also found at 864 and 751 cm^−1^. In addition, a characteristic absorption band for the epoxy group of ESO located around 820 cm^−1^ was observed.

The C-H stretching vibrations of PLA/LG-g-PLMA were at 2994, 2944, and 2921 cm^−1^. As for ESO, they were at 2920 and 2850 cm^−1^. With the gradual incorporation of ESO, the C-H stretching vibrations of the composites slightly shifted to 2993, 2942, and 2922 cm^−1^, which might suggest some intermolecular miscibility and interactions. Changes in the intensity of these peaks were noted, with the intensity of the absorption peak located at 2922 cm^−1^ gradually weakening or even becoming not obvious, and the absorption peak at 2942 cm^−1^ slightly intensified. In the infrared spectra of PLA/LG-g-PLMA/ESO, the epoxy group at 820 cm^−1^ disappeared, which demonstrated an interaction between the epoxy group of ESO and PLA/LG-g-PLMA [34,42,43].

The interaction between PLA/LG-g-PLMA and ESO could be caused by the possible hydrogen bonding between the terminal carbonyl group in PLA and the epoxy group in ESO [44]. As can be seen from the FT-IR of PLLA/ESO in Figure 1b, the C-H stretching vibrations of PLA appeared at 2997 and 2943 cm^−1^, and the C-H stretching vibrations of ESO appeared at 2920 and 2850 cm^−1^. With the addition of ESO, the C-H stretching vibrations of PLLA/ESO shifted to 2993, 2941, and 2922 cm^−1^ and the absorption peak at 2850 cm^−1^ intensified, and the disappearance of the epoxy group at 820 cm^−1^ was also observed, which may indicated the interaction and miscibility between the epoxy group of ESO and PLA [42]. Polar groups in plasticizers (e.g., epoxy groups) can enhance mechanical properties and are crucial for ideal compatibility [42]. Therefore, the added ESO may interact with PLA through hydrogen bonding, giving raise to the mechanical properties and the compatibility of composites. This analysis will be further confirmed by mechanical properties test and micromorphological observation.

### 3.2. Thermal Properties

The thermal property and crystalline behavior of the composites were analyzed by DSC and the results were shown in Figure 2 and Table 2. The pure PLA (PLLA) and PLA/LG-g-PLMA composite (PLLA1) showed a *T_g_* of 61.38 °C and 60.08 °C, respectively. The incorporation of ESO lowered the glass transition temperature by 1–4 °C (PLLA2-PLLA6), which could have been caused by the plasticizing effect of ESO. This is associated with an increase in chain mobility due to penetration of plasticizer molecules into the PLA matrix with the addition of ESO as a plasticizer [42]. A similar phenomenon has been reported for biodegradable polymers using epoxidized vegetable oils as plasticizers [45,46]. The *T_cc_* of PLA /LG-g-PLMA/ESO composites shifted to a lower temperature (103.79 °C) as compared with PLA /LG-g-PLMA (107.85 °C), indicating that the addition of ESO enhanced the cold crystallization capacity of the composite. As shown in Figure 2a, the low-temperature melting peak (*T_m_*_1_) was less visible with the increase of ESO addition, which is considered to be indicative of a stable and flawless crystal [47]. However, this improvement in crystallization had only a minor impact on crystallinity (*X_c_*), which remained almost unchanged.

### 3.3. Thermal Stability

The TGA and DTG curves of the composites and the associated thermal property data are displayed in Figure 3 and Table 3. PLA/LG-g-PLMA composites started to decompose at 338.7 °C (*T*_5%_) with a maximum decompose temperature of about 373.8 °C (*T*_max_), which was similar to the *T*_5%_ and *T*_max_ of pure PLA. When ESO was added, the *T_5%_* and *T*_max_ of the composite shifted to a lower temperature. Higher loading of ESO (7%, and 9%) resulted in lower thermal stability of the composites. Similar finding have been reported by Zhao [48] and Thakur [49]. Two reasons may be accounted for this observation. One is that this decomposition behavior can be correlated with the plasticization effect of ESO, i.e., an increased mobility of PLA chains relative to one another at lower temperatures. The other is ESO, which, in the form of droplets in the composites, were uniformly dispersed in the composites at low additions. At high ESO loadings, the droplets aggregated and became larger in size, which decreased the compatibility with PLA, thereby leading to a reduction in thermal stability. Nevertheless, with an initial decomposition temperature still much higher than its processing temperature, the composites still presented good thermal stability during the melting processing.

### 3.4. Rheological Properties

Rotational rheometer testing is commonly used to evaluate interfacial interaction between the additives and polymer matrix. The storage modulus (*G*′), loss modulus (*G*″), complex viscosity (*η**), and loss factor (tan*δ*) versus angular frequency (*ω*) of composites were as shown in Figure 4. As can be seen from Figure 4, with the increase of frequency, all G′, G″, and η* of the composites with ESO were increased compared to the composites without ESO. The increase in *G*′ demonstrated enhanced elastic strength of composites, while the increased *G′*′ indicated reduced mobility of molecular chain [50].

As the ESO content increased, the η* increased and then decreased. Upon the addition of a small amount of ESO, the η* increased, and the composites showed solid-like behavior. It was related to the epoxy groups of ESO reacting with the carboxyl end group of PLA to form a crosslinked network, which entangled molecular chains and restricted the movement of chains. At higher loadings, ESO served as a plasticizer, and improved the movability of composites, as shown by the decrease in η*.

### 3.5. Mechanical Properties

The stress-strain curves of the composite films were shown in Figure 5a, and the associated tensile results were listed in Table 4. As shown in Table 4, PLLA/LG-g-PLMA had a tensile strength of 42.6 MPa, which gradually reduced to 26.5 MPa with the incorporation of ESO. The reason for this decrease in tensile strength may be twofold. On the one hand, the epoxy groups of ESO interacted with the terminal carboxyl group of PLA to form a crosslinked network, which reduced the rigidity of PLA. On the other hand, ESO also acted as a plasticizer for PLA, which softened the PLA chains. With the incorporation of ESO, the elongation at break and tensile toughness of PLLA/LG-g-PLMA/ESO were significantly enhanced, achieving maxima of 104.6% and 44.7 MJ/m^3^ at 5% loading, which were 18 and 11 times higher compared with those of the composites without the ESO addition (5.6%, 4.1 MJ/m^3^). At higher ESO loadings (7%, and 9%), the elongation at break and toughness began to decrease, down to 72.6% and 29.3 MJ/m^3^, but was still higher than that of PLLA/LG-g-PLMA and PLLA. When a small amount of ESO was added, the highly crosslinked network in the composites enhanced the capacity of energy dissipation, which together with the plasticization effect of ESO, resulted in a significant increasing in elongation at break up to 104.6%. However, with too high loading, ESO aggregated and increased in size, which would worsen or even destroy the compatibility and weaken the capacity of energy dissipation, leading to a lower elongation at break.

### 3.6. Morhology Analysis

SEM is essential for cryo-fractured surface morphology observation. As shown in Figure 6a, the surface of PLA/LG-g-PLMA was relatively smooth, and some cavities were observed. When a small amount of ESO was added, the phase-interface distinction was not obviously observed, indicating that the interfacial adhesion and compatibility of composites has been improved [30]. It is related to the reaction of epoxy groups of ESO with the terminal carboxyl group of PLA, which enhanced interfacial adhesion and interface structure, as shown in Figure 2. Compared with the smooth PLLA/LG-g-PLMA surface, the roughness increased, and the cavities size decreased after ESO addition, indicating an improvement in toughness. At 3% and 5% ESO addition, partial plastic deformation (red circles) can be observed in Figure 6c,d, indicating that the toughness of composities have been substantially improved. With the continuous addition of ESO, the cavities size increased obviously, giving rise to negative influence on toughness.

The tensile fractured surface morphologies of PLA/LG-g-PLMA/ESO were observed by SEM (Figure 7), aiming to reveal the toughening effect of ESO in the composites. Without ESO addition, some twisted wrinkled on the tensile fracture surface of PLA/LG-g-PLMA were observed, indicating a fast relaxation process of the composite during the tensile test. With the addition of 1% ESO, a significant amount of plastic deformation was observed for the tensile fracture surface, which contributed to the fracture energy dissipation during tension [24,51]. At higher ESO loadings, significant elongation of the matrix at the point of plastic deformation can be observed, indicating substantial increase in the toughness of the composites [51,52]. When the tensile stress is higher than the interfacial adhesion between the phases, the particle will be debonded from the PLA matrix, forming cavities and inducing shear yielding, which results in large matrix plastic deformation [48]. Therefore, it was proposed that the tensile toughening mechanism of composites follows the internal cavitation-induced matrix shear yielding, which provides a remarkably effective way to dissipate energy. However, as shown in Figure 7e,f, the size of the cavities in PLA/LG-g-PLMA composites with 7% and 9% ESO were too large, which were not stable enough during the tensile test, and therefore tended to break before matrix shear yielding. This was consistent with the trend of elongation at break of the composites.

### 3.7. Optical Properties

UV energy may affect food or food packaging components, exposing them to the risk of degradation and sensory changes. PLA is prone to oxidative degradation under UV radiation. In such cases, the UV-blocking function of food packaging material becomes critical [53]. Among UV, light UV-A has the lowest energy and longest wavelength, while UV-B is the most high-energy constituent of natural UV and is known to degrade polymer materials through photochemical processes.

Figure 8 showed the UV-Vis transmittance spectra of PLA/LG-g-PLMA composite films before and after the addition of ESO. Our previous research found that that LG-g-PLMA had provided the composites with excellent UV barrier properties [37]. As can be seen from Figure 8, PLA was highly transparent, with absorption only in the 200–250 nm region, while the composites were dark brown in color, and the addition of ESO barely affected the UV barrier properties of the composites, which still maintained excellent UV barrier properties. In the UV region of 200–400 nm, the transmittance was close to 0, indicating that the material is almost completely UV-blocking, and it can be considered that the composites have potential applications in the UV barrier packaging application.

### 3.8. Gas Barrier Properties

The oxygen barrier properties of PLA/LG-g-PLMA composite films before and after the addition of ESO were characterized by the measurement of oxygen permeability, and the results are shown in Figure 9. PLA/LG-g-PLMA/ESO composites exhibited an increase in oxygen permeability and thus a decrease in barrier properties. The introduced ESO served as a plasticizer, which provided PLA chains more mobility, thus increased the diffusion of oxygen molecules, and resulted in a reduction of the gas barrier properties of the composite films [33]. For composites containing ESO, the oxygen barrier was relatively low, especially for ESO content of 5 phr, with an oxygen permeability value of about 2.420 × 10^−14^ cm^3^ cm/(cm^2^ s Pa), which is still suitable for packaging some food products. Therefore, the composite films can be considered for packaging food products that are not prone to immediate oxidation, such as vegetables, fruits, salads, and bakery products. For oxygen-sensitive foods and dry food products that need to be hermetically sealed, such as fresh meat and potato chips, the composites can be aluminized to increase their oxygen barrier properties.

### 3.9. Overall Migration

Components in food packaging materials might migrate into food, and overall migration test is an efficient way to measure the non-volatile substances mass value of food contact materials migrating to food. An overall migration test was conducted for PLA/LG-g-PLMA/ESO films to evaluate its applicability for food contact applications. According to standard, water, 3% acetic acid, 10 vol % ethanol, 50 vol % ethanol, and 95 vol % ethanol aqueous solution were selected as the water simulant, aqueous food simulant, and fatty food simulant. The test results were shown in Figure 10.

Figure 10 showed that the overall migration of composite films increased with the addition of ESO and ethanol concentration of the simulants. However, the values were still below the overall migration limit (60 mg/kg simulant [18]), indicating that PLA/LG-g-PLMA/ESO composite films are safe for the packaging of water, hydrophilic and lipophilic liquids, and foodstuffs, such as drinkable water, peeled vegetables, peeled fruits, salads, bakery products, potato chips, and fresh meat.

## 4. Conclusions

In this work, in an effort to improve the compatibility and toughness of PLA/LG-g-PLMA composites, we introduced ESO as a plasticizer, which resulted in composite films with excellent toughness and elongation at break, providing a new strategy for environmentally friendly food packaging application.

The results of the thermal properties, mechanical properties, rheological properties, and fracture surface morphology showed that the toughening mechanism of ESO could be mainly attributed to (1) the plasticizing effect of ESO and (2) compatibility improvement due to a crosslinked network formation via the interaction between epoxy groups of ESO and the terminal carboxyl groups of PLA. ESO addition elevated the elongation at break of the composite from 5.6% to 104.6% and the toughness from 4.1 MJ/m^3^ to 44.7 MJ/m^3^. The composites achieved a transition from brittle to ductile.

The overall migration value of PLA/LG-g-PLMA/ESO composite films was lower than the permitted limit of 10 mg/dm^2^ for food contact use. The films also exhibited excellent UV barrier properties. Despite the decrease in oxygen barrier due to ESO introduction, the composite films could still be considered for packaging foods such as peeled fruits, peeled vegetables, salads, and bakery products.

## Data Availability

The original contributions presented in the study are included in the article, further inquiries can be directed to the corresponding author.

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
