# Peer review of "Epoxidized Soybean Oil Toughened Poly(lactic acid)/Lignin-g-Poly(lauryl methacrylate) Bio-Composite Films with Potential Food Packaging Application"

_polymers, 2024, doi:10.3390/polym16142025_

Round 1

Reviewer 1 Report

Comments and Suggestions for Authors

Overall, this is a very detailed presentation of Data and methods of a technologically interesting material, as biobased materials gain increasing interest in materials science and technology. So this paper is another contribution to investigate and optimize PLA blends with bio-derived fillers and modifiers and continues the research of various groups on the system PLA/LG-g-PLMA with another oil-based plasticizer. 

The introduction delivers a good survey of previously published investigations. For this investigation, I’d prefer to see a more detailed explanation of the chemical interactions of ESO not only with PLA-endgroups, but also with lignin- and methacrylate-functionalities on page 3.

The data of all synthetic and analytical methods are well presented and quite promising for future applications, but I’d advice the authors to add an analytical investigation of the extracts of the overall migration tests. This could enlighten possible reactions between the various components. And although the overall migration remains below the limits of the China National Standard it is important to know what are the extracted compounds. In addition, I’d like to see tests with food simulants based on other solvents than ehtanol like Food simulant B 3% acetic acid (w/v) or Food simulant D2, which is any vegetable oil containing less than 1% unsaponifiable matter and is closer in its chemistry to oily liquids as 95% ethanol.

After these minor additions the paper should be published.

Reviewer 2 Report

Comments and Suggestions for Authors

The authors produced PLA/LG-g-PLMA composites plasticized with ESO. It's an interesting manuscript, especially for so-called green composites. However, before publication, some recommendations are necessary:

General comment: It is recommended to add the properties of pure PLA for comparative purposes. Add pair DSC, TG, Rheological, and UV-vis transmittance spectra;

>Page 2. Line 76-92. Please make the manuscript clear about the novelty. For the first time, the combined effect of ESO and LG-g-PLMA was investigated?

In the introduction, authors should also briefly address how PLA plasticization is being conducted in the literature, for example, using impact modifiers based on EVA, ABS, PCL, PBAT, AES, between others. Then, show the difference of this manuscript, using an ecologically based plasticizer.

>Materials. Please add the melt flow index of PLA; For lignin add: purity, pH, decomposition temperature, density;

The experiments must be detailed for other researchers to reproduce in other research. Some details must be added, such as:

>Page 3. “composites were prepared with an internal 120 mixer…………..”. State the type of rotor used in the internal mixer;

Preparation of PLA/ LG-g-PLMA/ESO Composite Film - Inform the cooling time and the load used during cooling;

Fourier Transform Infrared Spectroscopy (FT-IR) - Indicate whether the analysis was conducted using the ATR method to minimize the effect of thickness;

Differential Scanning Calorimetry (DSC) - Indicate the mass used and the gas flow in the equipment; Add the base reference data "ΔH100%PLLA=93.7J/g";

Mechanical Properties - Indicate the load used during the test. Did you use an extensometer?

Thermogravimetric Analysis - Indicate the mass used and the gas flow of the equipment. Why he used a rate of 20°C/min is unclear. Normal is a lower rate for greater accuracy, default is 10°C/min;

Rheological Properties - The rheology test needs more details, such as: equipped with parallel plate geometry (what diameter)? What atmosphere? What is the gap between the plates? What is the deformation within the region of

linear viscoelasticity?

Scanning Electron Microscopy - Were the samples coated with gold?

>Figure 1. Why did the authors not develop PLA/ESO to evaluate the effect of oil on the PLA matrix?

“epoxy group at 820 cm-1 disappeared, which demonstrated an interaction between the epoxy group of ESO and….”. Authors must report other works in the literature with similar results from these interactions;

>Figure 2. Please review DSC equation 2. The graph shows the presence of cold crystallization, but the authors did not reduce the effect of cold crystallization in the equation;

> Thermal Stability. Please add the TG of neat PLA, ESO and LG-g-PLMA in Figure 3, so we will have a better idea of ​​individual behavior;

>Page 8. Mechanical Properties. The mechanical results were good. Please add the elastic modulus value for each composition. This way, we will have a better idea of ​​the effect of ESO on stiffness;

Comments on the Quality of English Language

Minor editing of English language required

Round 2

Reviewer 2 Report

Comments and Suggestions for Authors

The authors commented satisfactorily on the questions. Furthermore, recommendations were added to the manuscript, improving technical and scientific quality. Therefore, the new version of the manuscript has merit for publication.

Comments on the Quality of English Language

Minor editing of English language required